# Genome-Wide Identification, Characterization and Expression Analysis of Mango (*Mangifera indica* L.) chalcone synthase (*CHS*) Genes in Response to Light

Haofeng Hu [1,2,3,†], Bin Shi [1,2,3,†], Wencan Zhu [1,2,3], Bin Zheng [4], Kaibing Zhou [1,2,3], Minjie Qian [1,2,3,*,‡] and Hongxia Wu [4,*]

1   Sanya Nanfan Research Institute of Hainan University, Sanya 572025, China
2   School of Horticulture, Hainan University, Haikou 570228, China
3   Key Laboratory of Quality Regulation of Tropical Horticultural Crop in Hainan Province, Haikou 570228, China
4   Ministry of Agriculture Key Laboratory of Tropical Fruit Biology, South Subtropical Crops Research Institute, Chinese Academy of Tropical Agricultural Sciences, Zhanjiang 524013, China
*   Correspondence: minjie.qian@hainanu.edu.cn (M.Q.); wuhongxia@catas.cn (H.W.)
†   These authors contributed equally to this work.
‡   First correspondence.

**Abstract:** Light is one the most important environmental factors regulating the accumulation of specific metabolites in plants, such as flavonoid. Chalcone synthase (CHS) is the key rate-limiting enzyme of the flavonoid biosynthesis pathway, while little is known about the *CHS* genes in mango (*Mangifera indica* L.). Therefore, the aim of the study was to obtain the genome-wide identification of the *CHS* genes in mango and screen the light-responsive family members. In this study, a total of 21 *CHS* genes were identified from the mango genome, and most of the members contained the Cys-His-Asn triad and the CHS/STS signature motif. Most of the *MiCHS* genes were located on chromosomes 2 and 4, and seven pairs of tandem duplication genes and five pairs of segmental duplication genes were detected among the *MiCHS* members. The expression of *MiCHSs* showed a tissue-specific pattern and was not correlated with the flavonoid or anthocyanin accumulation in cultivars with different peel colors. *MiCHS4*, *MiCHS16* and *MiCHS17* were found to respond to preharvest and postharvest UV-B plus visible light treatments, which exhibited no significant relationship with the light-responsive cis-acting element distribution in the promoter region. Our results are helpful and can be used to further study the function of *MiCHS* genes in light-induced flavonoid biosynthesis in mango.

**Keywords:** mango; chalcone synthase; genome-wide; gene expression; light-responsive





## 1. Introduction

Sunlight is not only the predominant energy source of almost all the photosynthetic organisms but is also an important factor regulating diverse processes of plant growth and development, including seed germination [1,2], de-etiolation [3,4], flower induction [5] and shade avoidance [6,7]. Sunlight comprises of a continuous spectrum ranging from infrared to ultraviolet-B (UV-B), which is sensed by plants through different photoreceptors, i.e., phytochromes (red/far-red), cryptochromes, phototropins and Zeitlupe (blue/ UV-A), and UV Resistance Locus 8 (UVR8 for sensing UV-B) [8]. Most of the solar UV-B (280–315 nm) is absorbed by the ozone layer, accounting for less than 0.5% of the total light energy reaching the earth's surface [8]. Meanwhile, UV-B also functions as an important environmental signal, which can induce evident changes in the gene expression, metabolism and morphology in plants [9–11]. In general, genes contributing to plant defense are up-regulated by UV radiation, including genes participating in the repair of

DNA damage [12], the scavenging of reactive oxygen species (ROS) [13] and the synthesis of anti-oxidative and UV-B-absorbing metabolites, such as flavonoid [11].

The synthesis of flavonoid starts with the general phenylpropanoid pathway [14]. First, phenylalanine is catalyzed by phenylalanine ammonia-lyase (PAL), cinnamic acid 4-hydroxylase (C4H) and 4-coumarate-CoA ligase (4CL), respectively, to generate 4-coumaryl-CoA. Subsequently, the flavonoid biosynthesis pathway enters into the key step, which is also the rate-limiting step [15]. Chalcone synthase (CHS, E.C. 2.3.1.74), the gatekeeping enzyme, catalyzes the Claisen condensation reaction of one molecule of 4-coumaroyl-CoA and three molecules of malonyl-CoA to generate one molecule of naringenin chalcone, which is the precursor of various flavonoids [16]. CHS is known as the representative enzyme of plant-specific type III polyketide synthases (PKS), and according to the catalytic mechanism, PKS is classified into type I, II, and III [17]. Type I and II polyketide synthases exist in bacteria and fungi, while type III is plant-specific [18]. CHS is characterized by its possession of a Cys-His-Asn catalytic triad and two functional conserved domains, which are Chal_sti_synt_N (PF00195) and Chal_sti_synt_C (PF02797) [19,20]. There are many *PKS* super family genes, known as *CHS*-like genes, including genes encoding stilbene synthase (STS), 2-pyrone synthase (2-PS), benzalacetone synthase, biphenyl synthase, phlorisovalerophenone synthase, *p*-coumaroyltriacetic acid synthase, aloesone synthase, hexaketide synthase, pentaketide chromone synthase, etc. [21–27]. CHS-like genes differ from CHS in many aspects, such as their initial substrates, intermediate products and catalytic mechanisms, etc. [16]. Due to its crucial role in flavonoid biosynthesis, the genome-wide identification of *CHS* genes has been carried out for various plant species, including Arabidopsis [28], maize [20], rice [29], soybean [30], flax [31], moss [32], petunia [33] and eggplant [34]. Thus far, much information on the *CHS* gene family in mango is still unknown.

It is well established that *CHS* gene expression is induced by both UV and visible lights in numerous plants. *CHS* expression, therefore, is an excellent system that has been used to study the molecular mechanism of the regulated transcription of photoreceptors in Arabidopsis [35], maize [36], parsley [37,38], mustard [39,40] and French bean [41]. Most of the results showed that the HY5 binding site (ACE element or G-box) and MYB recognition element (MRE) are essential for the light response of the *CHS* gene [11,35,41].

Mango (*Mangifera indica* L.) is a representative plant of the Anacardiaceae family and one of the most popular tropical fruits in the world. The domestication of mango started in the Indo-Burmese and Southeast Asia regions 4000 years ago, and mango began spreading to other parts of the world in the fourteenth century [42,43]. Mango accounts for the fifth-largest fruit production in the world (http://www.fao.org/faostat/, accessed on 1 May 2022), and the world mango production accounted for 56 million tons in 2019 (https://www.fruitrop.com/, accessed on 1 May 2022). Mango has undergone whole-genome duplication events (WGD) multiple times [44]. Since the catalytic products of CHS and its downstream products play essential roles in diverse plant physiological processes, the *CHS* gene family has retained and expanded tremendously due to WGD events in mango. Members of a large gene family might acquire different expression patterns and functions during the evolutionary and adaptive process under a low selective pressure [45]. In this study, we performed the genome-wide identification of mango *CHS* genes, followed by their structural characterization, the construction of a phylogenetic tree and a synteny analysis. Then, the expression patterns of *MiCHSs* in different tissues (mature leaf, bark, seed, root, flower, peel and flesh), different peel color cultivars (red, yellow, and green) and responses to light treatment (preharvest bagging treatment and postharvest UV-B/visible light treatment) were analyzed. The light-responsive cis-acting elements in the promoter region of *MiCHSs* were analyzed, and their relationships with the gene expression pattern were also analyzed. The results of our study can provide cues for the further genetic study of the regulation of the expression of light-dependent *MiCHSs* in mango.

## 2. Materials and Methods

### 2.1. Identification and Annotation of Mango CHS Genes

First, we downloaded the complete whole-genome files and GFFs (annotation information file) of mango from The National Genome Science Data Center (https://bigd. big.ac.cn/search?dbId=gwh&q=PRJCA002248, accessed on 5 May 2022) and derived all the protein sequences of mango using TBtools [46]. Subsequently, hidden Markov model (HMM) data on the chalcone synthase domains (PF00195 and PF02797) were downloaded from Pfam (https://pfam.xfam.org/, accessed on 5 May 2022), and an HMM search was performed to obtain candidate members of the mango CHS proteins. Furthermore, the NCBI blastp online tool (https://www.ncbi.nlm.nih.gov/, accessed on 5 May 2022) was used to compare all of the candidate protein sequences with the Swiss-prot database in order to screen out near-source genes (*CHS*-like genes). Candidate members with incomplete conserved domains were recognized by the Batch CD-Seach online tools of NCBI and PfamScan (https://www.ebi.ac.uk/Tools/pfa/pfamscan/, accessed on 5 May 2022). After eliminating the candidates with incomplete domains, the redundant putative members of the mango CHS proteins were also discarded. Finally, all the mango MiCHS members were identified. The theoretical isoelectric point and molecular weight of the MiCHSs were predicted using Expasy (https://web.expasy.org/compute_pi/, accessed on 5 May 2022).

### 2.2. Sequence Alignment and Phylogenetic Tree Construction

The MsCHS2 of alfalfa (*Medicago sativa*) was used for the alignment with all the amino acid sequences of putative MiCHS proteins in MEGA-11 by muscle, and the parameters were set as default, except for the max iterations, being set to 1000. The Find Best DNA/Protein Models (ML) search in MEGA-11 was conducted with all the protein sequences of the MiCHSs, and the Gaps/Missing Data Treatment option was set as Partial Deletion 95%. Subsequently, the 'JTT + G' model was chosen to construct the maximum likelihood phylogenetic tree of the mango CHSs, bootstrap was set as '1000' and the Gaps/Missing Data Treatment was set as Partial Deletion 95%.

### 2.3. Excavation of Motifs and Conserved Domains in MiCHSs

The protein sequences of MiCHSs were uploaded to MEME (https://meme-suite. org/meme/tools/meme, accessed on 6 May 2022) in order to search for up to 10 motifs (Supplementary File S1). Two methods were conducted to excavate the conserved domains. Firstly, we used the Batch CD-Search online tools of NCBI (https://www.ncbi.nlm.nih. gov/, accessed on 6 May 2022) to identify whether the sequences contained complete conserved domains. Secondly, to obtain detailed information about the MiCHS domains, we performed a domain search on PfamScan (https://www.ebi.ac.uk/Tools/pfa/pfamscan/, accessed on 6 May 2022), with the expectation value set to '$1e^{-5}$'.

### 2.4. Gene Location and Duplication Analysis

The chromosome location and duplication analysis of the *MiCHSs* were performed using TBtools according to the annotation file of the mango genome (GFF). The MCScanX was conducted to information about the derive tandem duplications and segmental duplications among the *MiCHS* genes [47].

### 2.5. Materials and Treatments

All the mango cultivars used in this study, including 'Guifei', 'Jinhuang', 'Qingmang', 'Sensation' and 'Hongmang NO. 6', were harvested at the South Subtropical Crops Research Institute (SSCRI) in Zhanjiang, China.

Three cultivars with different skin colors, including 'Guifei' (red), 'Jinhuang' (yellow), and 'Qingmang' (green), were used to analyze the *MiCHS* expression. Three mature trees per cultivar were selected, with one tree per replicate. A total of 8 ripe fruits per tree were harvested, and the fruit peel was sampled for analysis.

The mango cultivar 'Sensation' was used for the bagging treatment. Three mature trees were chosen as the three biological replicates. A total of 50 fruits per tree were bagged in double-layered bags (Qingdao Kobayashi Co., Ltd., Qingdao, China) for light blocking at 20 days after full bloom (DAFB). The rest of the fruits exposed to sunlight served as the control. A total of 10 bagged and 10 control fruits per tree were harvested at 50, 80 and 120 DAFB, and the fruit peel was collected in liquid nitrogen and stored at $-80$ °C for further analysis.

The mango cultivar 'Hongmang NO. 6' was used for the postharvest UV-B/visible light treatment. Three mature trees were selected, and 90 fruits per tree were bagged at 20 DAFB, as described above. Green mature fruits were harvested, placed in bags and transported to the lab for treatment. The UV-B/visible light exposure was conducted in plant growth chambers (Conviron, Adaptis A 1000, Winnipeg, Canada), and UV-B light of a 4.5 $\mu W \cdot cm^{-2}$ intensity was generated by 1 narrowband UV lamp (PHILIPS PL-S 9W/01, 311 nm, Amsterdam, Holland). Visible light of an intensity around 16 $W \cdot m^{-2}$ was generated by 12 fluorescent tubes (Guangdong PAK Lighting Technology Co., Ltd., 28W/T5, Guangzhou, China). The temperature and humidity were set to 17 °C and 80%, respectively. In total, 180 unblemished fruits were divided into two groups (UV-B/visible light treatment and dark), each of which was further divided into three biological replicates, with 30 fruits per replicate. The peels of 5 fruits were sampled after 0, 6, 24, 72, 144 and 240 h of UV-B/visible light exposure, and the control sample (dark) was collected parallel.

*2.6. Measurement of Flavonoid and Anthocyanin Contents in Mango Peel*

The total flavonoid and anthocyanin contents were measured according to the protocol described by Pirie et al. [48]. The flavonoid and anthocyanin of 0.15 g of fruit peel were extracted using 1 mL methanol (with 0.01% HCl) and stored for 12 h at 4 °C in darkness. The resulting solution was centrifuged for 10 min at 12,000 rpm and 4 °C. A total of 200 μL of supernatant was measured using a microplate reader (Nano Quant, infinite M200, Tecan, Switzerland) at wavelengths of 325 and 530 nm for the detection of the total flavonoid and anthocyanin contents, respectively.

*2.7. Transcriptome Analysis of the MiCHS Expression*

The expression of the *MiCHSs* in different tissues, including the mature leaf, bark, seed, root, flower, peel and flesh, was analyzed using the transcriptome data of 'Alphonso' derived from the NCBI database [44]. The accession numbers are as follows: SRX7706076 (mature leaf), SRX7706065 (bark), SRX7706053 (seed), SRX7706052 (root), SRX7706044 (peel), SRX7706038 (flower) and SRX7706037 (flesh). The clean reads were obtained using the Fastp software (https://github.com/OpenGene/fastp, accessed on 10 September 2022) after the low-quality data of the raw reads were removed and subsequently mapped to the mango genome database (BIG Genome Sequence Archive database, accession number: PRJCA002248) by TopHat [49]. The transcripts were assembled from the clean reads using Cufflinks, and the gene expression was presented as fragments per kilobase of transcript per million fragments mapped (FPKM) = mapped fragments of transcript / [total count of mapped fragments (millions) × length of transcript (kb)] [49].

*2.8. Analysis of Quantitative Real-Time PCR*

The RNA Prep Pure Plant Kit, designed specifically for plant tissues rich in polysaccharides and polyphenolics (Tiangen, DP441, Beijing, China), was used to extract RNA from 0.1 g of fruit peel, and the genomic DNA was digested using DNaseI. A NanoDropLite spectrophotometer (Thermo Scientific, Waltham, USA) was used to determine the concentration of total RNA. HiScript IIQ RT SuperMix (Vazyme, R223-01, Nanjing, China) was used to the reverse transcription of cDNA with 1 μg of RNA. Then, 20-times diluted cDNA was used as the template for the quantitative real-time PCR (Q-PCR) analysis. The Q-PCR reaction solution (total volume 15 μL) consisted of 7.5 μL SYBR premix ExTaqTMII (Takara, Japan), 1 μL of each primer (10 μM) and 5.5 μL of cDNA. We performed the

reaction using a real-time PCR machine (qTOWER3G, Jena, Germany) starting at 95 °C for 30 s and then continuing with 40 cycles of 95 °C for 5 s and 60 °C for 30 s. For the Q-PCR primers, first, we designed the primers for 21 *MiCHS* genes, ran Q-PCR with the fruit peel samples and performed the melting curve analysis. In particular, for *MiCHS13* and *MiCHS14*, which showed a high sequence similarity and could not be distinguished by the primers, primers used to amplify the same product were chosen. Then, 11 *MiCHS* genes were found to be expressed in the fruit peel according to the transcriptome data and Q-PCR. The primers used to amplify products showing a unique peak of the melting curve were finally chosen for Q-PCR. All the Q-PCR primers are listed in Supplementary File S2. The $2^{-\Delta\Delta Ct}$ method and mango *actin* gene were chosen to calculate the gene expression and normalization, respectively.

### 2.9. Promoter Cis-Acting Element Analysis

Next, 2000 bp upstream genome sequences from the start codon 'ATG' of the *MiCHS* genes were considered as the candidate promoters for the light-responsive cis-acting element analysis. The HY5 binding site (ACE element or G-box) and MYB recognition element (MRE) were retained. All the cis-acting elements were analyzed using PlantCARE (http://bioinformatics.psb.ugent.be/webtools/plantcare/html/, accessed on 6 May 2022).

### 2.10. Statistical Analysis

Experimental data were subjected to a one-way analysis of variance (ANOVA), with the mean values separated by Tukey's multiple range test using SPSS 27.0 (SPSS, Chicago, IL, USA). Probability values of $<0.05$ were considered statistically significant.

## 3. Results

### 3.1. A Total of 21 Non-Redundant Putative Members of the MiCHS Gene Family Were Identified

In order to identify the candidate protein sequences of the MiCHSs in the whole genome of the mango, the HMM data of Chal_sti_synt_N (PF00195) and Chal_sti_synt_C (PF02797) were downloaded on Pfam and an HMM Search was run using TBtools. In total, 63 candidates were obtained by the first step and, subsequently, all the amino acid sequences of the 63 candidates were matched with all the reference species protein sequences in the Swiss-prot database on NCBI. After screening out the near-source genes (*CHS-like* genes), 25 candidate members were left. A Batch CD-Search and a PfamScan were performed using the 25 candidates to eliminate the candidates with incomplete domains. At the end, 21 non-redundant putative members of the *MiCHS* genes were identified and named *MiCHS1* to *MiCHS21* in the order of their gene IDs (Table 1). The amino acid sequence lengths of these MiCHS proteins varied from 303 to 411, and 13 of them (61.9%) had around 390 amino acids (Table 1). Furthermore, the predicted molecular weights and theoretical pIs of these MiCHSs varied from 33 to 46 kDa and 5.97 to 8.74, respectively (Table 1).

**Table 1.** Details of putative *MiCHS* genes.

| Gene Name | Gene ID | Chr ID | Length (aa) | pI | Mw (Da) |
|-----------|---------|--------|-------------|-----|---------|
| *MiCHS1* | mango002786.t1 | chr 2 | 391 | 6.5 | 42,943.65 |
| *MiCHS2* | mango002798.t1 | chr 2 | 391 | 6.53 | 42,647.35 |
| *MiCHS3* | mango002799.t1 | chr 2 | 304 | 6.09 | 33,475.56 |
| *MiCHS4* | mango002800.t1 | chr 2 | 391 | 6.18 | 42,700.39 |
| *MiCHS5* | mango002801.t1 | chr 2 | 391 | 6.18 | 42,754.52 |
| *MiCHS6* | mango002802.t1 | chr 2 | 390 | 5.97 | 42,756.57 |
| *MiCHS7* | mango002803.t1 | chr 2 | 390 | 5.97 | 42,845.66 |
| *MiCHS8* | mango002804.t1 | chr 2 | 391 | 7.15 | 42,882.67 |
| *MiCHS9* | mango003116.t1 | chr 2 | 411 | 6.44 | 45,379.59 |
| *MiCHS10* | mango007035.t1 | chr 4 | 411 | 6.38 | 46,069.54 |
| *MiCHS11* | mango007036.t1 | chr 4 | 354 | 6.41 | 39,076.37 |

**Table 1.** *Cont.*

| Gene Name | Gene ID | Chr ID | Length (aa) | pI | Mw (Da) |
|---|---|---|---|---|---|
| *MiCHS12* | mango007038.t1 | chr 4 | 391 | 6.12 | 42,672.46 |
| *MiCHS13* | mango007039.t1 | chr 4 | 391 | 6.22 | 42,735.56 |
| *MiCHS14* | mango007040.t1 | chr 4 | 367 | 6.72 | 39,992.42 |
| *MiCHS15* | mango007048.t1 | chr 4 | 347 | 6.38 | 38,199.12 |
| *MiCHS16* | mango018655.t1 | chr 11 | 398 | 6.03 | 43,295.06 |
| *MiCHS17* | mango020263.t1 | chr 12 | 390 | 8.74 | 42,118.31 |
| *MiCHS18* | mango021008.t1 | chr 12 | 390 | 6.67 | 42,755.25 |
| *MiCHS19* | mango021009.t1 | chr 12 | 344 | 6.41 | 37,407.22 |
| *MiCHS20* | mango023633.t1 | chr 14 | 391 | 6.13 | 42,833.43 |
| *MiCHS21* | mango025592.t1 | chr 15 | 391 | 6.42 | 42,755.48 |

Chr: chromosome; pI: theoretical isoelectric point; Mw: molecular weight (average).

### 3.2. Multiple Sequence Alignment of Putative MiCHS Proteins

Ferrer et al. obtained the crystal structure of MsCHS2 from alfalfa (*Medicago sativa*) and revealed that four active site residues (Cys 164, Phe 215, His 303 and Asn 336), especially the Cys-His-Asn triad, are critical for the substrate binding of MsCHS2 [16]. To determine whether the putative MiCHS sequences contain the active site residues and CHS/STS signature motif (WGVLFGFGPGLT), a multiple sequence alignment was established using the 21 deduced amino acid sequences of the putative mango CHS proteins with MsCHS2. The four active site residues (Cys 164, Phe 215, His 303 and Asn 336) were conserved in all the MiCHS proteins, except for MiCHS11 and MiCHS15, which had a deletion in the middle of the sequences, resulting in a lack of Phe 215 (Figure 1). Except for MiCHS3 and MiCHS10, which had a deletion and a fragment substitution at the end of the sequences, respectively, leading to the loss of the CHS/STS signature motif, the motif was conserved in all the MiCHS proteins (Figure 1). It is commonly observed that the fragment deletion, insertion or substitution occurrs during the evolution of genes. For instance, MiCHS9 showed an insertion at the N terminal, MiCHS14 exhibited a fragmental deletion in the middle of the sequence, and a fragment deletion and a fragment substitution were detected at the C terminal of MiCHS3 and MiCHS10, respectively (Figure 1).

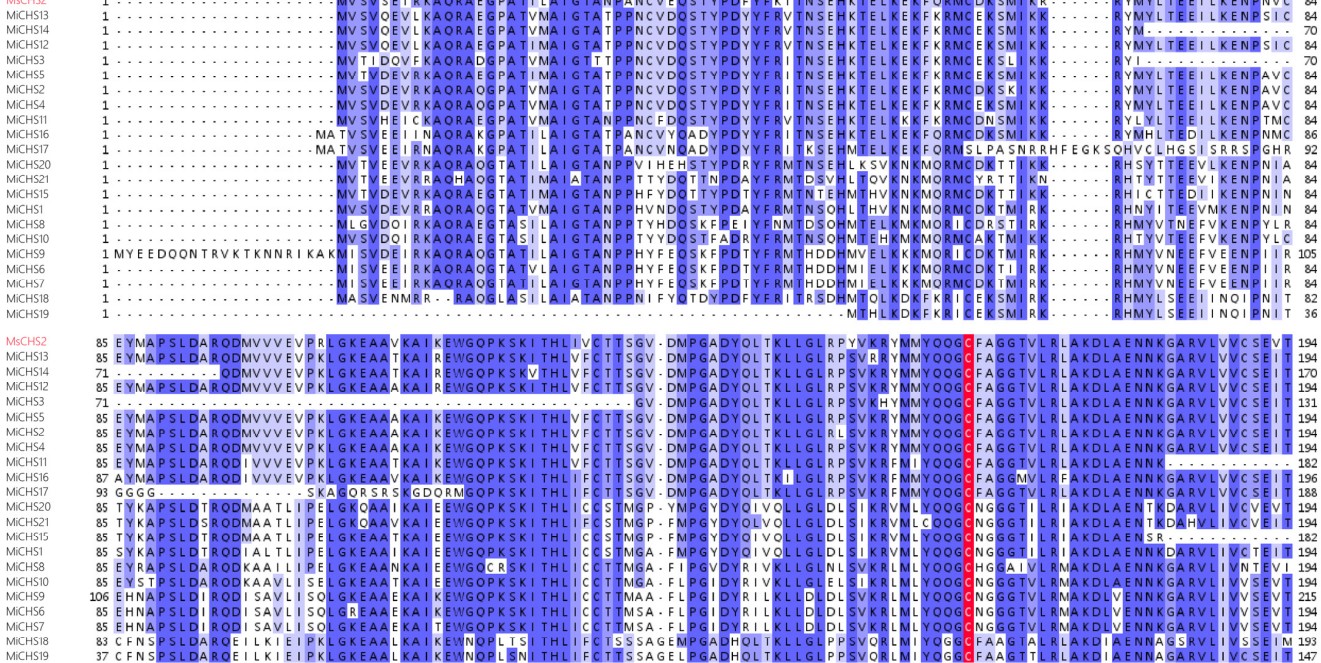

**Figure 1.** *Cont.*

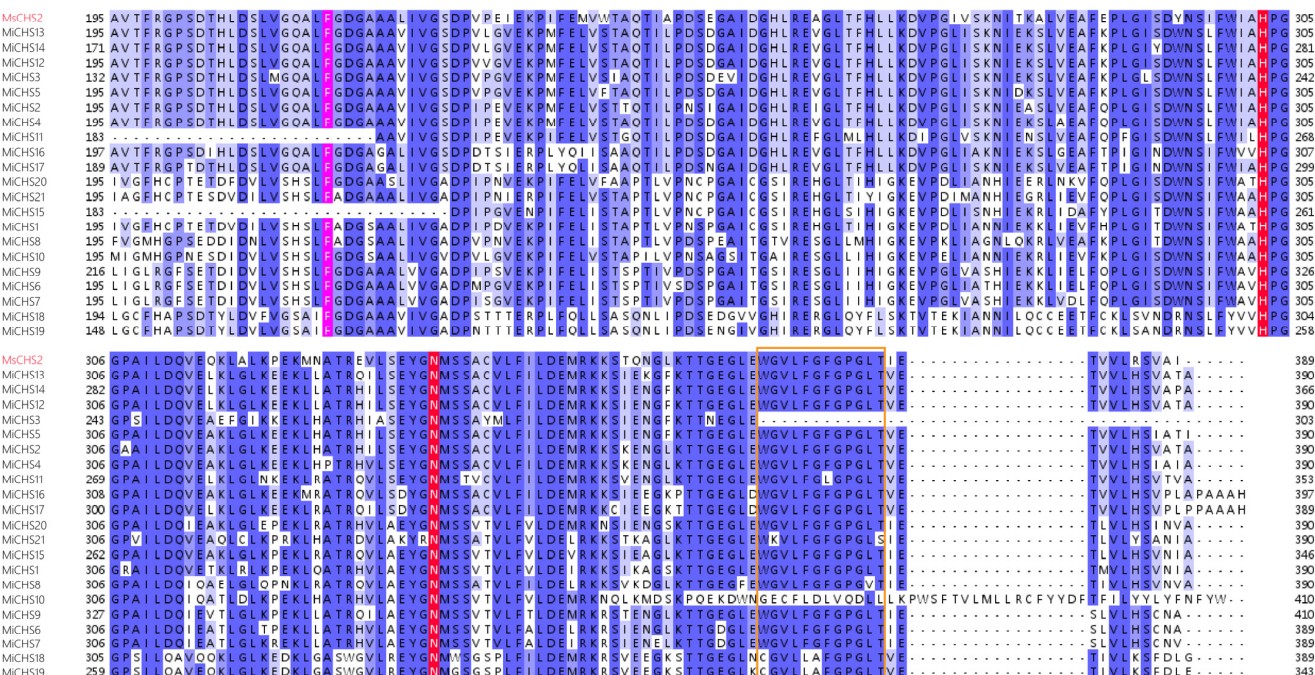

**Figure 1.** Multiple sequence alignment of the deduced peptides of MiCHS, together with MsCHS2 (P30074), from *Medicago sativa*. The CHS/STS signature motifs (WGVLFGFGPGLT) are framed by an orange box. The active site Cys-His-Asn triad and Phe based on MsCHS2 are highlighted in red and purple, respectively. The darker blue indicates higher numbers of identical residues. Gaps are represented by short hyphens.

### 3.3. Construction of the Phylogenetic Tree, Motif Pattern, Domain and Gene Structure

A maximum likelihood tree was constructed, along with the motif pattern, domain information and gene structure, for further phylogenetic analysis using the amino acid sequences of the putative MiCHS proteins (Figure 2). The MiCHSs were classified into four distinct clades according to their phylogenetic relationships, and the largest subfamily was clade 1, containing 10 members of MiCHS, followed by clade 3 (5 members), clade 2 (4 members) and clade 4 (2 members) (Figure 2a). The motif distribution pattern of the MiCHS proteins was essentially identical between the different members. A total of 14 out of 21 members contained all 10 motifs, and all the members contained the chalcone synthase signature motif (motif 7), except for MiCHS3 and MiCHS10 (Figure 2b). The MiCHSs in clade 4 did not contain motif 9, and members in clade 3 showed an identical motif distribution pattern containing all the motifs (Figure 2b). Motifs 1, 3, 6 and 10 were present0 in all the members, and motif 5 showed the lowest occurrence rate (18/21) and was absent in MiCHS3, MiCHS14 and MiCHS17 (Figure 2b).

The gene structure of *MiCHSs* revealed two different patterns of the CDS structure, and most of the *MiCHS* genes contained two exons and one intron (Figure 2c). However, *MiCHS3*, *MiCHS11*, *MiCHS14*, *MiCHS15*, *MiCHS18* and *MiCHS19* were composed of three exons and two introns (Figure 2c). The genomic sequences of *MiCHS5* and *MiCHS13* contained only one long intron each, which were about 4000 and 15,000 bp, respectively (Figure 2c). The Chal_sti_synt_N domain was generally composed of the first exon and part of the second exon, and the Chal_sti_synt_C domain was distributed in the remaining part of the second exon (Figure 2c).

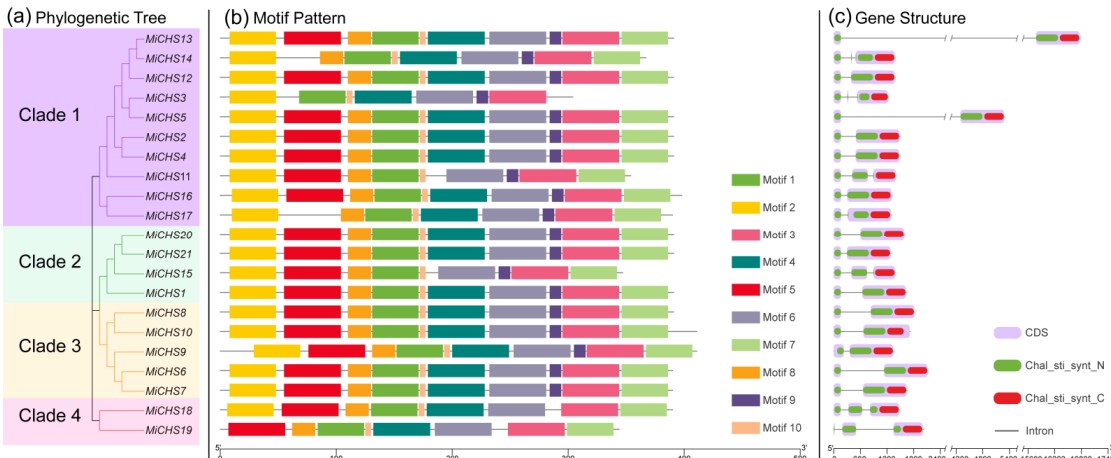

**Figure 2.** Phylogenetic tree, motif pattern and gene structure of *CHS* genes from the mango. (**a**) Based on the complete sequence of mango CHS proteins, a phylogenetic tree was constructed using MEGA-11 software. Different colors are used to represent different clades. (**b**) Composition of the mango CHS protein motifs. Patterns numbered 1 to 10 appear in differently colored boxes. Sequence information for each subject is supplied in Supplementary File S1. The protein length can be estimated using the following scales. (**c**) Gene structure of mango *CHSs*. Purple boxes denote coding sequences (CDS). The green and red regions represent the conserved domains of the MiCHS protein N-terminal and C-terminal, respectively, and black lines indicate introns.

### 3.4. Synteny Analysis and Chromosomal Location of MiCHSs

The physical locations of the *MiCHS* genes on chromosomes and duplication information are shown in Figure 3. The *MiCHS* genes were distributed unevenly across chromosomes 2, 4, 11, 12, 14 and 15, and most of them formed tight clusters. Chromosomes 2, 4 and 12 contained nine, six and three members of the *MiCHS* genes, respectively, while only one member of the *MiCHS* was detected on chromosomes 11, 14 and 15. Furthermore, seven pairs of tandem duplications were found, including *MiCHS2* and *MiCHS3*, *MiCHS4* and *MiCHS5*, *MiCHS6* and *MiCHS7*, *MiCHS7* and *MiCHS8*, *MiCHS12* and *MiCHS13*, *MiCHS13* and *MiCHS14* and *MiCHS18* and *MiCHS19*. In addition, five pairs of segmental duplications, including *MiCHS1* and *MiCHS15*, *MiCHS2* and *MiCHS12*, *MiCHS4* and *MiCHS11*, *MiCHS7* and *MiCHS10* and *MiCHS16* and *MiCHS17*, were detected. Except for *MiCHS16* and *MiCHS17*, the remaining four pairs of segmental duplications were reverse parallel (Supplementary File S3).

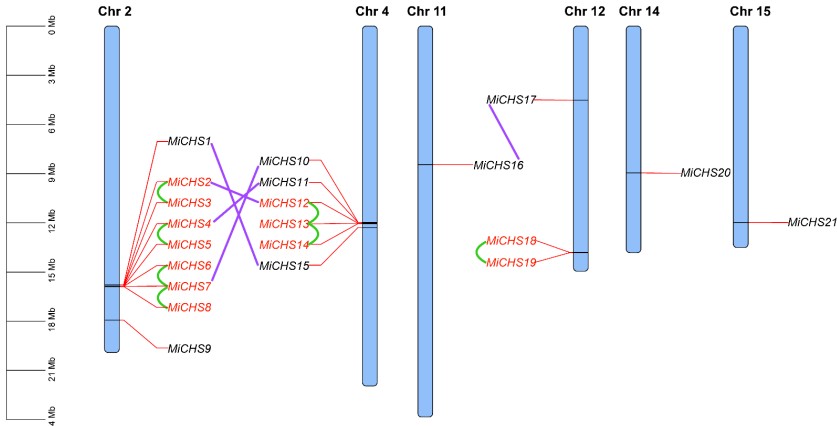

**Figure 3.** Chromosome distribution analysis of mango *CHS* gene family. A total of 12 *MiCHS* genes marked in red font and paired with green arcs are defined as tandem duplication events. Purple lines indicate segmental duplications of the *MiCHS* genes.

### 3.5. Tissue-Specific Expression Pattern Analysis of MiCHS Genes

The transcriptome data of 'Alphonso', including the mature bark, flesh, flower, leaf, peel and root, were downloaded for the analysis of the tissue-specific expression pattern of 21 *MiCHS* genes. As shown in Figure 4, the *MiCHS* genes belonging to clade 1 showed a higher expression level than clades 2, 3 and 4. Compared with the other members, *MiCHS16* showed the highest expression in the bark, flower, leaf, root and seed. The transcriptions of *MiCHS2*, *MiCHS4*, *MiCHS5*, *MiCHS11*, *MiCHS16* and *MiCHS17* exhibited a similar pattern, with the greatest abundance in the root and seed, followed by the flower, bark, peel and leaf. These genes were clustered closely on the phylogenetic tree. *MiCHS7*, *MiCHS9* and *MiCHS20* exhibited a relatively high expression level only in the root, while the expression of *MiCHS21* could not be detected in any tissue. All the *MiCHS* genes showed no expression or a very low expression level in the flesh. Interestingly, high expression levels of *MiCHS2*, *MiCHS4*, *MiCHS5*, *MiCHS16* and *MiCHS19* were detected in the fruit peel. The FPKM of all 21 *MiCHS* genes in the mango organs/tissues with the transcriptome data can be found in Supplementary File S5.

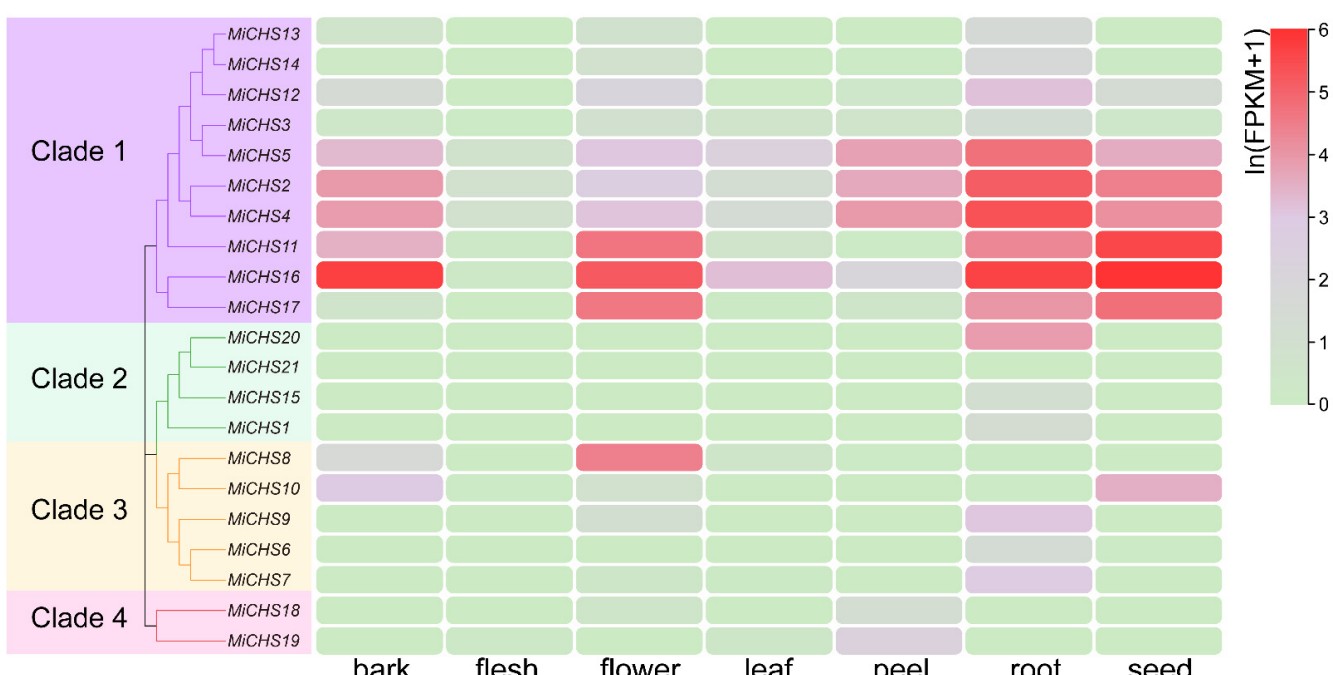

**Figure 4.** Hierarchical clustering of mango *CHS* expression profiles in different tissues. The 21 *MiCHS* genes are divided into Clades 1–4. The color of the bar represents the expression value.

### 3.6. Flavonoid and Anthocyanin Contents and MiCHS Expression in Differentially Colored Mango Cultivars

Three different mango cultivars with distinct peel colors, including 'Guifei' (red), 'Jinhuang' (yellow) and 'Qingmang' (green), were used to detect the flavonoid and antho-cyanin contents and expression of the *MiCHS* genes (Figure 5a). The red cultivar 'Guifei' showed by far the highest contents of flavonoid and anthocyanin, while 'Jinhuang' and 'Qingmang' showed no difference between one another (Figure 5b). Based on the transcrip-tome data and primary Q-PCR analysis, 11 *MiCHS* genes were expressed in the peel of mango, including *MiCHS2*, *MiCHS4*, *MiCHS5*, *MiCHS8*, *MiCHS11*, *MiCHS12*, *MiCHS13/14* (which could not be distinguished by the primers), *MiCHS16*, *MiCHS17* and *MiCHS19*. Most of the *MiCHS* genes, including *MiCHS2*, *MiCHS4*, *MiCHS5*, *MiCHS11*, *MiCHS12* and *MiCHS13/14*, were abundant in 'Qingmang' compared with 'Guifei' and 'Jinhuang' (Figure 5c). In contrast, *MiCHS16* showed the highest expression level in 'Jinhuang' and

no difference in the *MiCHS17* expression could be detected between the three cultivars (Figure 5c). *MiCHS8* was not expressed in the ripe fruit peel of the three cultivars (Figure 5c).

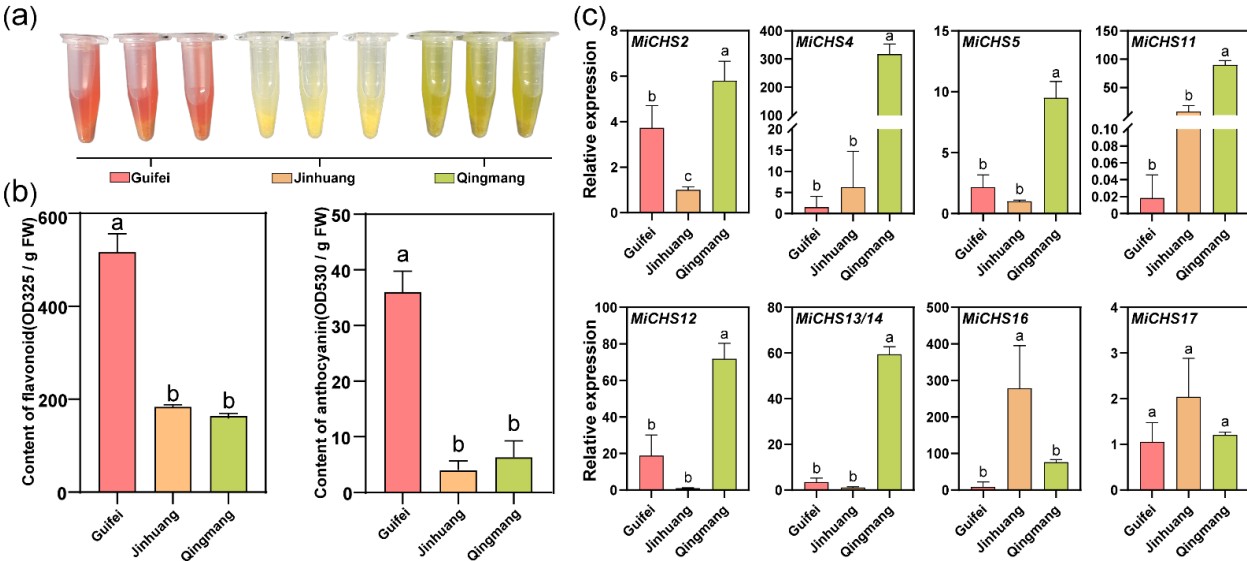

**Figure 5.** (**a**) Extraction solutions for flavonoid and anthocyanin in 'Guifei', 'Jinhuang' and 'Qingmang' ripe fruit peels. (**b**) Flavonoid and anthocyanin contents in the ripe fruit peels of three mango cultivars. (**c**) *MiCHS* genes expression levels in 'Guifei', 'Jinhuang' and 'Qingmang' fruit peels. The data represent the mean value ± SD, *n* = 3. Different letters above the bar chart indicate a significant difference at *p* < 0.05.

### 3.7. Expression of MiCHS Genes in Response to Light

During the bagging treatment, the expressions of *MiCHS4*, *MiCHS16* and *MiCHS17* were significantly induced by sunlight, with a higher transcription level in the control sample compared with the bagged samples at the most stages of development (Figure 6). In contrast, the expression of *MiCHS5* was inhibited by sunlight at 80 DAFB (Figure 6). The expression of the rest of the genes did not show a strong response to sunlight (Figure 6). In addition, *MiCHS8* was only expressed at 50 DAFB, while *MiCHS11*, *MiCHS12* and *MiCHS19* were expressed at the later developmental stages (Figure 6).

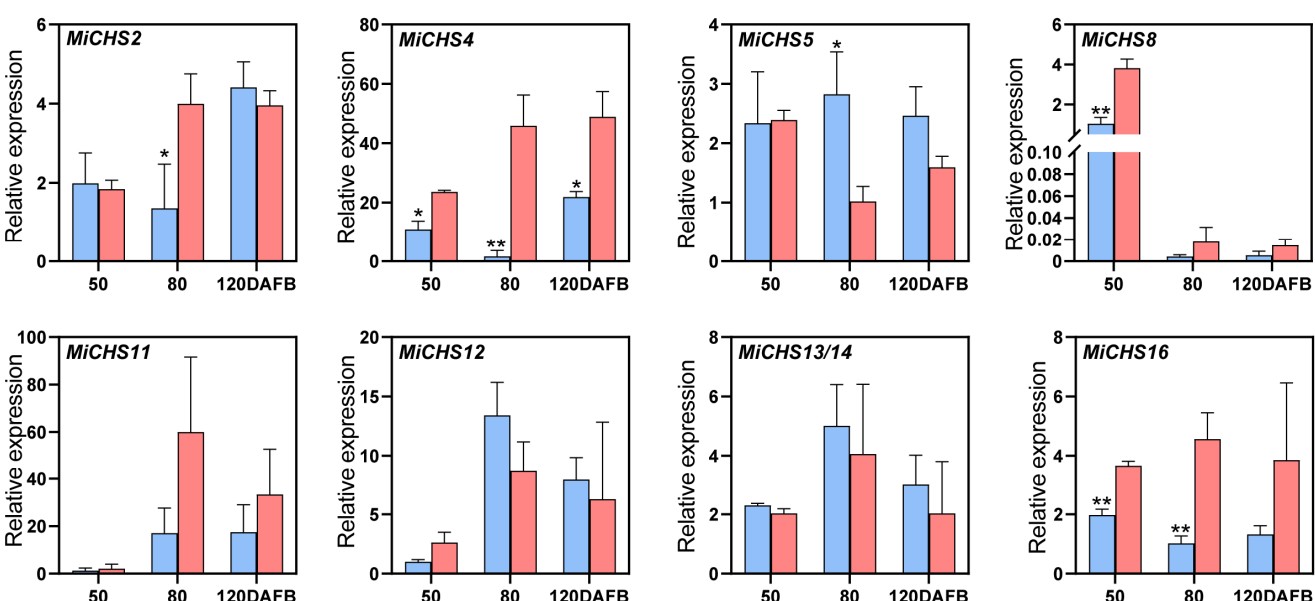

**Figure 6.** *Cont.*

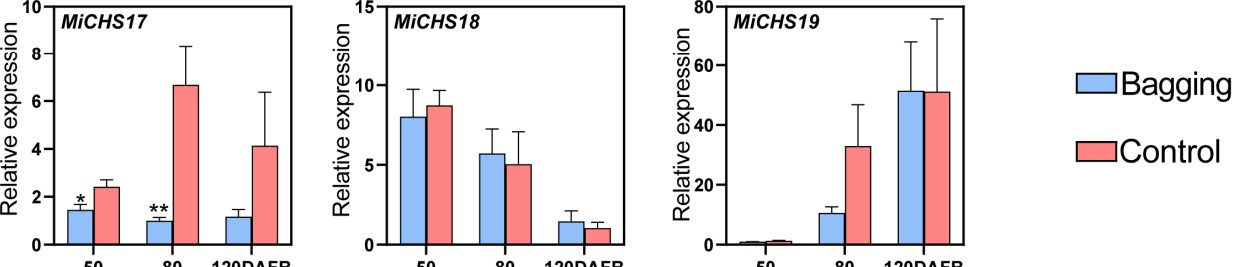

**Figure 6.** Expression profiles of *MiCHS* genes during the bagging treatment. Bagged (blue bars) and control (red bars) 'Sensation' mango peels were collected at 50, 80 and 120 DAFB. The data represent the mean value ± SD, *n* = 3. * indicates a significant difference (*p*-value < 0.05). ** indicates a very significant difference (*p*-value < 0.01), as determined by Student's *t*-test.

During the postharvest UV-B/visible light treatment, the transcriptions of *MiCHS4*, *MiCHS16* and *MiCHS17* were also up-regulated, while the response pattern was different (Figure 7). The response of *MiCHS17* to light lasted for almost the whole period of treatment (Figure 7). The responses of *MiCHS4* and *MiCHS16* to light occurred at the middle and late stages (144H and 240H) and early and late stages (6H, 24H and 240H) of light treatment, respectively (Figure 7). The expression of *MiCHS11* was also induced by light at 24H and 240H, while the rest of the genes did not an show obvious responsive pattern to the postharvest UV-B/visible light treatment (Figure 7).

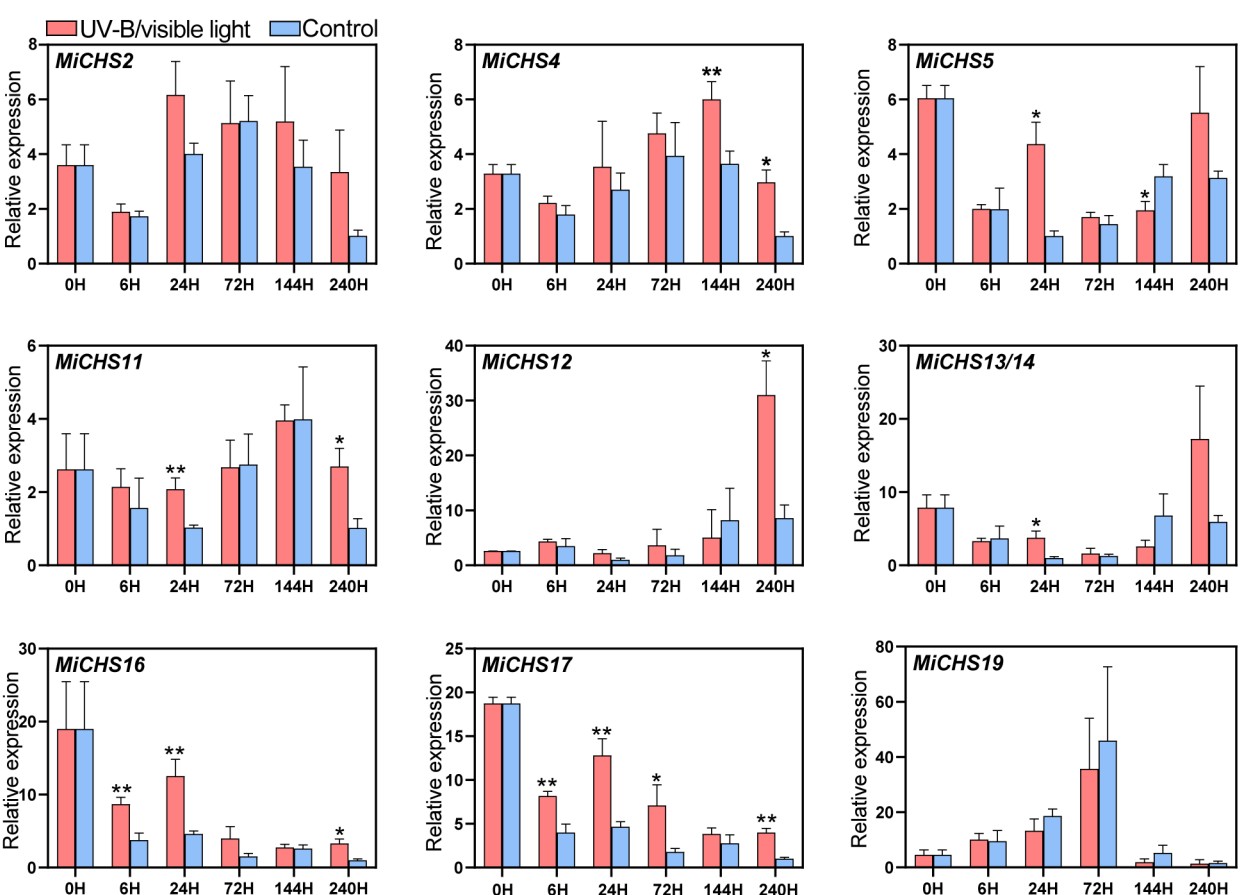

**Figure 7.** The expression of *CHS* genes in 'Hongmang NO. 6' mango peel was analyzed using a UV-B/visible light treatment and control (dark). The data represent the mean value ± SD, *n* = 3. * indicates a significant difference (*p*-value < 0.05). ** indicates a very significant differences (*p*-value < 0.01), as determined by Student's *t*-test.

Combining the two light treatments together, *MiCHS4*, *MiCHS16* and *MiCHS17* were defined as light-responsive *CHSs* in mango.

### 3.8. Analysis of Light-Responsive Cis-Acting Elements in the Promoter Region of MiCHS Genes

It has been reported that cis-acting elements, including the HY5 binding site (G-box or ACE element) and MYB recognition element (MRE), are crucial for the response of *CHS* to light. Therefore, 2000 bp upstream sequences of the mature fruit peel expressing 11 *MiCHS* genes were defined as the promoters and submitted to PlantCARE in order to search for the light-responsive elements. *MiCHS4*, *MiCHS16* and *MiCHS17* contained one G-box and four MREs, one G-box and five MREs, and six G-boxes, respectively (Figure 8). The other genes, which did not show an obvious response to light, exhibited a divergent pattern in the distribution of elements. For instance, *MiCHS5* and *MiCHS11* showed numerous elements, containing three G-boxes, one ACE, and three MREs, and four G-boxes and three MREs, respectively, while *MiCHS12* and *MiCHS18* only contained two MREs, and one MRE, and one ACE, respectively (Figure 8).

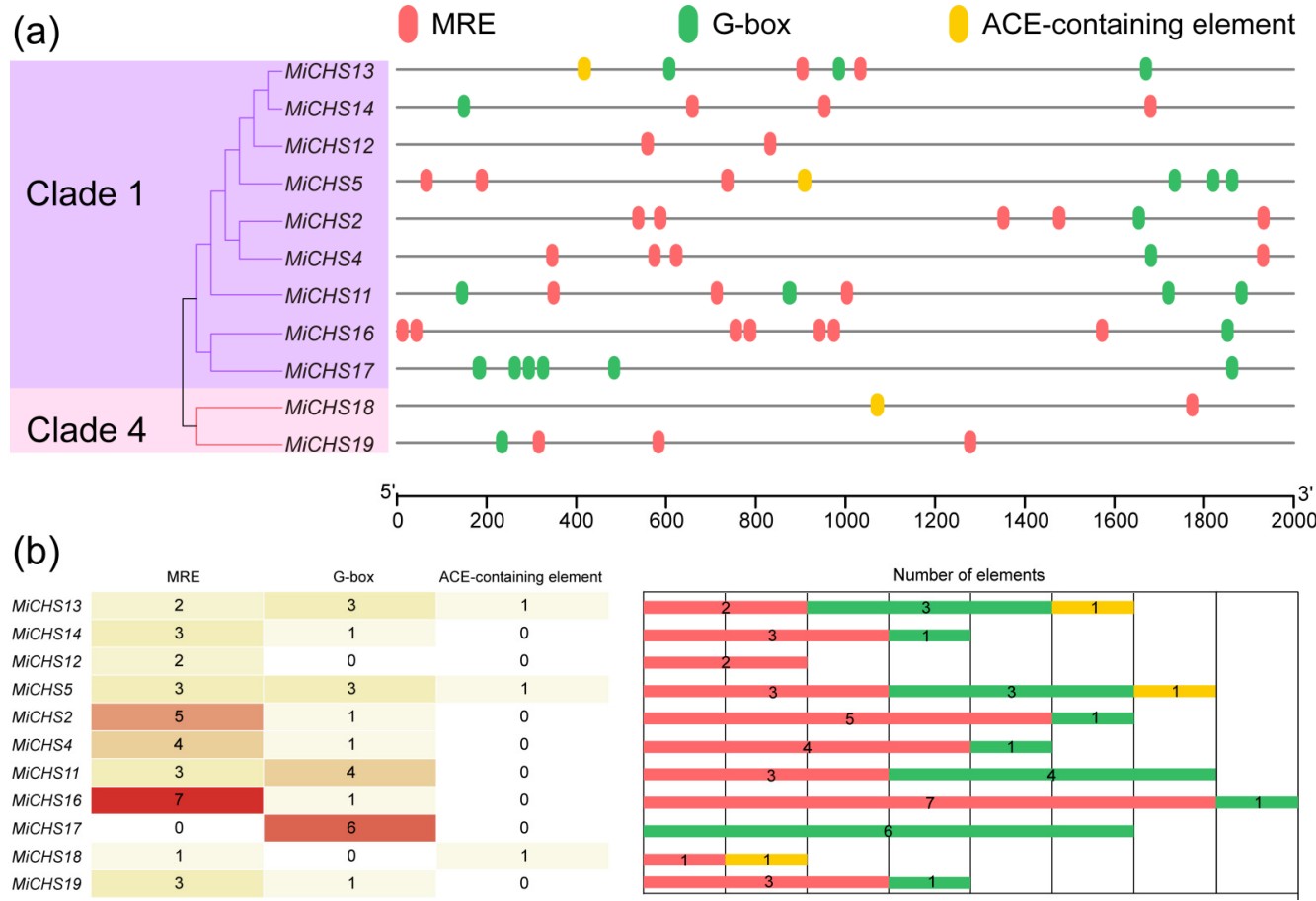

**Figure 8.** Analysis of light-responsive cis-acting elements in the promoter region of the 11 *MiCHS*. (**a**) The distribution of light-responsive cis-regulatory elements in the 2 kb upstream sequences of 11 *MiCHS* genes were analyzed using PlantCARE. (**b**) The statistics of light-responsive cis-regulatory elements for 11 *MiCHS* genes.

### 4. Discussion

A gene family is the result of many factors, such as tandem duplication, segmental duplication, whole-genome duplication (WGD) events and domestication, etc. [50,51]. Chalcone synthase catalyzes 4-coumaroyl-CoA and malonyl-CoA to generate naringenin-chalcone, which is the precursor of multiple crucial downstream metabolites. *CHS* generally forms a gene family with several members in various plant species, except for *Arabidopsis*

*thaliana*, which contains merely one *CHS* in the whole genome [18,52]. Thus far, 14, 14, 7 and 27 *CHS* genes have been identified in maize (*Zea mays*), soybean (*Glycine max*), eggplant (*Solanum melongena*) and rice (*Oryza Sativa*), respectively [20,29,30,34]. In this study, a total of 21 members of the mango *CHS* gene family were identified, representing one of the largest *CHS* gene families in plants (Table 1). Mango has undergone multiple WGD events, and the most recent might date back to ~70 million years ago [44]. Seven pairs of tandem duplications and five pairs of segmental duplications were found among the *MiCHS* genes, with most genes located on chr 2 and chr 4 (Figure 3). Among the 14 maize *CHS* genes, two segmental duplication events were found, while no tandem duplication was detected, indicating that segmental duplication is the main contributor to the expansion of maize *CHS* genes [20]. Generally, tandem duplication generates new genes, while segmental duplication disperses the gene copies, resulting in the slow evolution of the genes [53]. Clearly, the maize *CHS* gene family is evolving slowly, while in our case, both tandem duplication and segmental duplication contribute greatly to the enormous expansion of the mango *CHS* gene family.

The crystal structure of MsCHS2 from *Medicago sativa* showed that the four amino acid residues, including Cys 164, Phe 215, His 303 and Asn 336, are the active sites, and the Cys-His-Asn triad is essential for the substrate binding of the CHS enzyme [16]. It has also been reported that Phe 215 is the 'gatekeeper' blocking the lower part of the opening between the active site cavity and the CoA binding tunnel [21]. Among the 21 *MiCHS* genes, all the genes contained the Cys-His-Asn triad, while Phe 215 was missing in MiCHS11 and MiCHS15 due to the deletions of the peptide sequences (Figure 1). In addition, all the MiCHS proteins contained the CHS/STS signature motif, except for MiCHS3 and MiCHS10, which showed a deletion and substitution at the C-terminal of the proteins, respectively (Figure 1). All these mutations suggest that the relevant *MiCHS* genes might have different substrate preferences and, consequently, have divergent functions.

It is well known that genes from the same family exhibit diverse temporal and spatial expression patterns, and *CHS* genes are usually expressed in a tissue-specific manner. Among the 14 soybean *CHS* genes, most of the genes were highly expressed in the leaves, and even the three copies of *GmCHS3* showed divergent expressions in soybean tissue [30]. In eggplant, seven *CHS* genes contributed to the responses of root, stem, leaf, flower and peel to heat stress in a tissue-specific pattern [34]. Among the 14 maize *CHS* genes, only *ZmCHS01/02* exhibited a relatively higher expression in the seeds and leaves, while the rest of the genes showed low transcription levels in most maize tissues [20]. In this study, most of the high-expression genes were from clade 1 (Figure 4), indicating a close relationship between the gene evolution and gene expression. The high expression levels of *MiCHS2*, *MiCHS4*, *MiCHS5*, *MiCHS16* and *MiCHS19* in fruit peel suggest the potential roles of these genes in the physiological processes occurring in the fruit peel, such as flavonoid accumulation and responses to environmental stimulations. In addition, most of the mango *CHS* genes were highly expressed in the root and seed, but they were not expressed in the mature leaf (Figure 4), possibly indicating higher flavonoid and auxin concentrations in the root and seed compared to the mature leaf. Flavonoid has been implicated in the regulation of auxin movements, and flavonoid is concentrated in the tissues with vigorous growth patterns, including the upper hypocotyl, the hypocotyl–root transition zone and the distal elongation region of the root in Arabidopsis seedlings, which are the sites of auxin accumulation [54]. Thus, we assume that the differential expressions of *MiCHS* genes in the root, seed and mature leaf are associated with the function of flavonoid in auxin movement.

Anthocyanin is a subgroup of flavonoids. Thus, three mango cultivars with different fruit peel colors (red, yellow, and green) were used to investigate the relationship between the *MiCHS* expression and flavonoid and anthocyanin accumulation. It has been reported that most anthocyanin biosynthetic and regulatory genes, including *CHS*, showed a higher expression in the red mango cultivar 'Janardhan Pasand' than the green cultivar 'Amrapali' and the yellow cultivar 'Arka Anmol', except for *DFR*, which showed the

highest expression in the yellow cultivar 'Arka Anmol' [55]. In the green-skinned pear (*Pyrus pyrifolia*) cultivar 'Zaosu' and its red sport 'Zaosu Red', only the expressions of the key structural gene *PpUFGT2* and regulatory gene *PpMYB10* were highly correlated with the anthocyanin concentration, while the expressions of four *PpCHS* genes showed a young tissue preference instead of a preference for anthocyanin [56]. In this study, the highest flavonoid and anthocyanin contents were detected in the red cultivar 'Guifei', while most of the *CHS* genes showed the highest expression in the green-skinned mango cultivar 'Qingmang' (Figure 5). All these results indicated that, as the first gene in the flavonoid pathway, the *CHS* expression is not always highly correlated with flavonoid or anthocyanin accumulation.

Light is an essential environmental factor regulating the expression of key flavonoid biosynthetic genes such as *CHS* and, subsequently, leads to the accumulation of flavonoid in plants. Meanwhile, the *CHS* family members showed differential expression patterns in response to light treatment. Among the four pear *CHS* genes, the expression of *PpCHS1* and *PpCHS2* was light-inducible under the conditions of a preharvest bagging treatment and postharvest UV-B/visible light treatment, while the *PpCHS4* expression was inhibited by light during the bag treatment [57]. In cucumber (*Cucumis sativus*), three *CHS* genes were identified from the genome, and only *CsCHS2* was strong induced by UV treatment, which was associated with the enrichment of the HY5 binding site (ACE element or G-box) and MYB recognition element (MRE) in the promoter region [11]. The importance of the HY5 and MYB binding sites for the response of *CHS* genes to light has been investigated in various plant species [35–40]. In this study, *MiCHS4*, *MiCHS16* and *MiCHS17* were defined as light-responsive *CHSs* in mango based on the bagging treatment and postharvest UV-B/visible light treatment (Figures 6 and 7). However, these three genes did not show an increasing number of HY5 and MYB binding sites in the promoter region compared with the other genes that were not regulated by light (Figure 8). There was only a G-box and no MRE motifs in the promoter of *MiCHS17*, which did not support the theory that both the HY5 and MYB binding sites are necessary for the induction of *CHS* genes by light [35]. All these results indicated that the mechanism of light-induced *CHS* expression in plants differs between species, and there might be a unique regulation system in the case of mango.

## 5. Conclusions

A total of 21 non-redundant *CHS genes* were identified from the mango genome, and these genes could be divided into four clades. All the CHS members contained the Cys-His-Asn triad, which is essential for the substrate binding of the CHS enzyme, while the CHS/STS signature motif was missing in MiCHS3 and MiCHS10 due to a deletion and substitution at the C-terminal of the proteins, respectively. The distribution of motifs and domains was highly conserved among the *CHS* genes. Most of the *CHS* genes (15 out of 21) were located in chromosomes 2 and 4, with seven pairs of tandem duplications and five pairs of segmental duplications among the family members. The expression of the *CHS* genes exhibited a tissue-specific pattern and showed no relationship with flavonoid or anthocyanin accumulation. *MiCHS4*, *MiCHS16* and *MiCHS17* were defined as light-responsive *CHSs* in mango, but their expression was not highly correlated with the HY5 and MYB binding sites in the promoter. Our results could contribute to the further investigation of the function of *CHS* genes in light-induced flavonoid biosynthesis in the mango.

**Supplementary Materials:** The following supporting information can be downloaded at: https://www.mdpi.com/article/10.3390/horticulturae8100968/s1, File S1: Analysis and distribution of conserved motifs in mango CHS proteins; File S2: Sequences of the primers used in this study; File S3: Segmentally and tandemly duplicated *MiCHS* gene pairs; File S4: Melting curve analysis of *CHS* and *actin* gene primers in three mango cultivars. File S5: Public RNA-seq data of 21 *MiCHS* genes that were used in this study.

**Author Contributions:** Conceptualization, H.H., B.S., M.Q. and H.W.; methodology, H.H., B.S., W.Z., B.Z. and K.Z.; data curation, H.H., B.S., H.W. and M.Q.; writing—original draft preparation, H.H., B.S., W.Z., B.Z., K.Z., M.Q. and H.W.; writing—review and editing, H.H., B.S., M.Q. and H.W.; funding acquisition, H.W. and M.Q. All authors have read and agreed to the published version of the manuscript.

**Funding:** This research was funded by the National Natural Science Foundation of China (grant number: 32160678), the Major Science and Technology Plan of Hainan Province (grant number: ZDKJ2021014), Hainan Provincial Natural Science Foundation of China (grant numbers: 322RC568; 320QN192), the National Key Research and Development Plan of China (grant number: 2018YFD1000 504) and the Scientific Research Foundation of Hainan University (grant number: KYQD(ZR)20053).

**Data Availability Statement:** Not applicable.

**Acknowledgments:** We thank Shuo Ma and Xueli Zhang from the Guangdong Provincial People's Hospital for their great help in the writing and revising of the paper.

**Conflicts of Interest:** The authors declare no conflict of interest.

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
