# Peer review of "Genome-Wide Identification, Characterization and Expression Analysis of Mango (Mangifera indica L.) chalcone synthase (CHS) Genes in Response to Light"

_horticulturae, doi:10.3390/horticulturae8100968_

Round 1

Reviewer 1 Report (Previous Reviewer 2)

Study entitled "Genome-wide identification, characterization and expression analysis of mango (Mangifera indica L.) chalcone synthase (CHS) genes in response to light" sounds interesting. 

But I have some major and minor concerns with this study.

Minor concerns

1. In introduction section details about the importance and significance of Manggifera indica L. is missing. Author focussed on on technique only. Provide details of origin and importance of selected organism along with interest of study.

2. In Material and method section 2.1 Author mentioned "Furthermore, NCBI blastp online tool (https://www.ncbi.nlm.nih.gov/) was used to compare all of the candidate protein sequences with Swiss-prot database in order to screen out near-source genes". Author should download Swiss-prot database followed by standalone blastp analysis. If so, clear this point.

Major concerns

1. As per the results of Figure 5, why most of the MiCHS genes are found to be highly expressed in root and seed and and no expression in leaf? Explain briefly.

2. I think author should explore more about protein-protein interaction networks, only mapping on String database followed by image provided by database is not enough. Either remove PPI from this study or explain it briefly with good discussions including effect of interactions on other pathways by using up or down regulatory signatures.

Author Response

Reviewer 2 Report (Previous Reviewer 1)

Major points
1. Fig. 7: Simultaneous application of UV-B and white light does not follow standard methodology of plant physiology. Low intensity of white light activates light signaling and promotes photosynthesis, while high
intensity of white light and UV-B act as environmental stress to suppress photosynthesis. Simultaneous application of UV-B and white light with a negative control of dark treatment complicates physiological situations. The authors should check effect of UV-B and white light individually. Therefore, additional analysis is required to replace Fig. 7.
 Thank you for your comments. Please let us interpret. We think you are meaning Figure 8, instead of Figure 7. Results of figure 7 and figure 8 were used to select the light responsive CHS genes. Figure 7 is based on preharvest bagging treatment, in which bagged fruits (darkness) were regarded as negative control and unbagged fruits (exposed to sunlight which contain both UV and white light) as light treatment. In Figure 8 posthavest light treatment, we tried to imitate the light conditions in figure 7, so we use the prehavest and postharvest light treatment with similar light quality to select the light responsive CHS genes. That’s why we use postharvest UV-B/white light treatment corresponding to ‘unbagged treatment’, and darkness corresponding to ‘bagging treatment’.
 As you mentioned, high intensity of white light and UV-B act as environmental stress. So in the current study, we use low white light (4000 Lux) and UV-B (4.5 μW•cm-2) does, which are much lower than the light stress level. The purpose of setting low does of light is we want to set light as an environmental signal instead of an environmental stress. The light responsive CHS genes we want to select is through the light transduction pathway instead of the stress (ROS, ect.). We think our experiment design didn’t complicates the situation.
 But we think your idea is very interesting. Probably in the future, we could also screen CHS genes responding to different light qualities, such as far-red, red, blue, UV, etc. For the current study, you could regard UV-B/white light as a whole part, sort of ‘mimic sunlight’.
We hope our explanation makes sense to you.

Reviewer 1: I mentioned Fig. 7 in the previous ms, which is Fig. 8 in the revised ms. 

Application of UV-B triggers stress response even in low dose, therefore it is wrong to say low dose application of UV-B should not give any stress response. It does. 

Anyway, the experimental conditions shown in the ms are not suitable to judge light responsiveness of the CHS genes. Legends of bars in the revised Fig. 8 ("UV-B" and "Control" ) are misleading. 

I noticed light intensity is not appropriately expressed in M & M. "Lux" (L156) should not be used in plant science papers. "W/m2" and "umol/m2/s" are good expressions.

Minor points

4. Abstract: The second sentence says CHS is the first enzyme of flavonoid biosynthesis pathway. I guess it is PAL, not CHS.
PAL is the first enzyme of phenylpropanoid pathway, which provide the substrate for devious secondary metabolites, such as flavonoid and lignin. So actually, CHS is regarded as the first specific enzyme of flavonoid pathway.

Reviewer 1: Hans Mohr says that phenylalanine is a starting substance of the biosynthetic pathway of flavonoids (Fig. 18.2, Mohr and Schopfer, Plant Physiology, Springer, 1995). If the authors want to respect Professor Mohr, the first enzyme should be PAL. CHS is sometimes referred to as the key enzymes of the flavonoid biosynthesis, which expression is  acceptable.

Author Response

Reviewer 3 Report (New Reviewer)

The manuscript describes the initial investigation of chalcone synthase genes in mango. The results provide an initial overview of this gene family in mango, and are a starting point for more detailed functional and genetic/genomic studies. 

In general, the manuscript is well written, with clear methods and results, and the discussion is logically based on the results. 

A minor point about the qPCR methods. The expression of 11 genes is shown in the results – were these chosen from the transcriptome analysis, or was the expression of all 21 genes also tested by qPCR? In lines 324-325, it is stated “Based on the transcriptome data and primary Q-PCR analysis…”. Some description of this in the methods would provide clarity, and the primer design process couls also be briefly described, particularly if the duplicated gene pairs were analysed by qPCR.

Minor text corrections:

Lines 84-85: “structural characterisation”

Line 87: “different peel color cultivars”

Line 90 and Line 201: change “excavated” to “identified” or similar

Line 103 and Line 214: please clarify “order to screen out near-source genes”. It is not clear to me what this means

The spelling of one of the cultivars is inconsistent - ‘Geifei’ or ‘Guifei’?

Line 213: “Swiss-prot”

Line 231: “deduced”

Line 233: “except”

In general, the text is understandable, however, language proofreading would correct minor language details that could improve the readability of the manuscript.

Round 2

Reviewer 2 Report (Previous Reviewer 1)

Expression of Fig. 7 is changed so as to avoid misunderstanding the results. Considering experimental limitation in the horticulture field, I recommend publication of the ms on MDPI Horticultuae with minor revision.

Minor points

*Abstract (L27): "light" should be changed to "UV-B plus visible light".

Author Response

*Abstract (L27): "light" should be changed to "UV-B plus visible light".

A:Correction has been made.

This manuscript is a resubmission of an earlier submission. The following is a list of the peer review reports and author responses from that submission.

Round 1

Reviewer 1 Report

Table 1, FIg. 1, 3 and 4 are useful for mango researchers, but the other data are not. Therefore, this work is premature and should include more data which helps mango researchers for publication.

Major points

*Comparison between cultivars should be deleted to give a good focus of the paper.

*Expression data of CHS should be compared with accumulation profiles of anthocyanin. There is a statement in the abstract saying no correlation between anthocyanin accumulation and the expression profile of CHS genes, but I could not find anthocyanin data in the figure and supplemental data.

*Please try to show results which lead to biological knowledge. Now I could not feel physiological significance of the presented data.

Minor points

*Fig. 2b have a large duplication with Fig. 1. It is better to change Motif 1 to 10 to established protein motifs in the protein databases.

*Fig. 8 should be deleted because the presented data is not useful. 

Reviewer 2 Report

The authors have conducted a study entitled “Genome-wide identification, characterization and expression analysis of mango (Mangifera indica L.) lhalcone synthase (CHS) gene in response to light” seems interesting. 

But there are few minor and major concerns with this study.

Minor concerns:- 

1. Introduction seems like discussion, First introduce targeted plant followed by its importance and current issues. 

Major Concerns:-

1. In Material and method Author mentioned about “Transcriptome analysis of MiCHSs expression” But accession number: PRJNA48715 

Mentioned by Author belongs to Streptococcus pneumoniae. How expressions were checked? Explain in detail with full methods and stat analysis. 

2. What are the different conditions and time gap for Transcriptome data expression.

3. How author has managed samples for real time PCR analysis if data is available in public platform and not generated by Author. If that data belongs to your group then mention all details in the manuscript.  

4. Why author ignored Protein-Protein interaction networks for CHS proteins, it might give more detailed information about closely associated proteins associated with metabolic pathways in a clear way followed by expression. 

Reviewer 3 Report

The manuscript entitled "Genome-wide identification, characterization and expression analysis of mango (Mangifera indica L.) chalcone synthase (CHS) genes in response to light" by Haofeng Hu and collogues looks interest and the content of this article is easy to understand. The results and discussion are clearly written. The conclusion is consistent with presented results that the expression of CHS genes are a tissue-specific manner, and have no relationship with anthocyanin accumulation. 

This study provides the basic information that might contribute for the further function investigation of CHS genes during the light-induced flavonoid biosynthesis in the mango. However, there are some points to address before publication:

1. This study focuses only the identification and expression of CHS genes. I can not see any analysis or result that relate to genome-wide analysis. Where are the integrative analyzes or results that infer genome-wide analysis? Otherwise, the authors should reconsider to the name of this study.

2. The Abstract has no objective.

3. In line 163-164, please elaborate more information regarding to the Transcriptome Data used in this study. Is this data already published. If yes, please cite reference.

4. In line 195 and throughout this manuscript, I suggest the authors edit "no-redundant" to "non-redundant".

5. Regarding to the statement in line 397-399 and overall content in this manuscript, what is the difference between the anthocyanin/flavonoid contents of these selected three mango varieties with different peel colors? Please provide the measured data of anthocyanin/flavonoid contents between the different peel colors and different mango varieties. In addition, statistical analysis should be apply to infer the difference between the anthocyanin/flavonoid contents of the studied samples.

Reviewer 4 Report

In this work, entitled “Genome-wide identification, characterization, and expression

analysis of mango (Mangifera indica L.) chalcone synthase (CHS) genes in response to light”, the authors identified CHS genes from the mango genome and characterized its genomic structure, expression profile, and promoter region. Additionally, the authors characterized the response of the CHS genes to light.

In general, the manuscript is very interesting and well documented.

I have a few comments and suggestions that are listed below.

Materials and methods:

Line 96: extracted all of the deduced proteins encoding for CHS from mango’s genome??? indicate it because it is confusing that is indicated that all proteins were extracted

In supplementary file S2, include the amplicon size and efficiency of each pair of primers

Discussion:

Line 404: Please, indicate the species studied of the PgGENES first cited in the text

Round 2

Reviewer 1 Report

This ms introduces studies on mango CHS genes which encode the key enzyme in flavonoid biosynthesis. Authors have identified 21 mango CHS genes from genome sequence, which is the first genome-wide identification in mango. Comparison of the amino acid sequences, chromosomal localization of the genes, and expression profiling regarding tissue specificity and also light response was done and provides useful information to mango researchers.

Major points

1. Fig. 7

Simultaneous application of UV-B and white light does not follow standard methodology of plant physiology. Low intensity of white light activates light signaling and promotes photosynthesis, while high intensity of white light and UV-B act as environmental stress to suppress photosynthesis. Simultaneous application of UV-B and white light with a negative control of dark treatment complicates physiological situations. The authors should check effect of UV-B and white light individually. Therefore, additional analysis is required to replace Fig. 7. 

In addition, genes for positive control of the treatments should be included. I mean, an UV-B responsive gene should be included in UV-B response analysis, and a light-responsive gene should be included in light response analysis.

Minor points

1. RT-PCR analysis (Fig. 5, 6, 7)

Include expression data of a control gene, like actin or ubiquitin, to show unbiased RNA extraction, quantitation, and cDNA synthesis. 

2. Cultivar analysis (Fig. 5)

Show success of RT-PCR by electrophoretogram of the product or melting curve which indicate single product by RT-PCR with the utilized primer sets. Show it in supplemental figures. I worry about sequence variations in the cultivars which does not fit with the utilized primers. I apologize if they are already included in supplemental data.

3. Abstract 

The second sentence says CHS is the first enzyme of flavonoid biosynthesis pathway. I guess it is PAL, not CHS. 

4. Fig. 5 

The legends of three cultivars in the top of the figure are not very necessary, because they are also explained in the bottom of each graph.

Reviewer 2 Report

BioProject ID provided by author consists of 105 transcriptome datasets.

It will be difficult for future researchers to find which specific data related to transcriptome was selected for analysis. Number varies from time to time in public platforms due to more sequencing projects related to organism.

Thus, data specific SRA accessions are required to regenerate results in future.

Just mapping proteins on already available Arabidopsis network and claiming  establishment of a protein-protein interaction network is not reliable. Use full (Interlog based or Orthrolog based) methodology for PPI construction and mention the details.